



# Medium-scale gravity waves observational methodology for antarctic airglow observations.

Gabriel Augusto Giongo[1], Cristiano Max Wrasse[1], Pierre-Dominique Pautet[2], José Valentin Bageston[3], Prosper Kwamla Nyassor[1], Cosme Alexandre Oliveira Barros Figueiredo[4], Anderson vestena Bilibio[1], Delano Gobbi[1], and Hisao Takahashi[1]

[1]Space Weather Division, National Institute for Space Research, Brazil.
[2]Physics Department, Utah State University, USA.
[3]Southern Space Coordination, National Institute for Space Research, Brazil.
[4]Universidade Federal de Campina Grande, Brazil.

**Correspondence:** Gabriel Augusto Giongo (gabriel.giongo@inpe.br)

**Abstract.** Medium-scale gravity waves are atmospheric waves with a horizontal scale of 50 to 1000 km. They can be observed via airglow all-sky images through the keogram technique. Our research introduces a novel algorithm that automatically identifies these waves, visible in airglow keograms, to study gravity waves over the Antarctic Peninsula. The all-sky airglow imager was installed at the Brazilian Comandante Ferraz Antarctic Station (EACF, 62°S), near the tip of the Antarctic Peninsula. The preprocessing techniques, including projection into geographical coordinates, flat fielding, consecutive image subtraction, and Butterworth filter, were used to enhance the visibility of the medium-scale waves. Based on the wavelet transform, the analysis procedure is used to identify the primary oscillation present in the keograms and reconstruct them posteriorly to check the wave coherency and obtain the wave parameters by fitting the phase lines in the phase domain. The fitting parameters then estimate wave parameters and the estimation quality. Simulations with synthetic images containing typical traveling waves were used to assess the error generated through the procedure and determine the fitting parameters threshold. This procedure was used to process a year of data in less than one hour, identifying most waves with errors below 5 percent. Waves observed parameters range close to the expected results, although they differ from other observation sites by having larger phase speeds and wavelengths.

## 1 Introduction

Atmospheric gravity waves are an essential feature of atmospheric motions due to their energy and momentum driving across the atmospheric layers (Nappo, 2002; Fritts and Alexander, 2003). Gravity waves over the Antarctic region transport momentum and energy through the atmosphere (Plougonven et al., 2013; Moffat-Griffin et al., 2011; Lu et al., 2015), playing a major role alongside other phenomena like the stratospheric polar vortex, Polar night jet (PNJ) and mesospheric polar clouds (Moffat-Griffin and Colwell, 2017; Zhao et al., 2015). They can also transport energy from one area of the planet to other regions due to their horizontal movement, influencing the dynamical features of the entire planet (Kogure et al., 2018; Vadas et al., 2019).



To understand the gravity waves' influence in the atmosphere, previous works have measured the gravity wave characteristics in order to parameterize their effects depending on their observed scales (Vosper et al., 2018; Lehner and Rotach, 2018). To do so, such observations must comprise large data sets and consider short- and long-term variations of the waves' characteristics as well as short-term variations to correctly associate them with environmental features and behavior (Song et al., 2021).

Instrumental biases are an essential aspect to be considered when observing gravity waves. An instrument's resolution constrains the dimensions of the waves it can observe (Wright et al., 2016, 2017). Spectral limitation and continuity of the data are also significant limitations to consider when observing traveling waves because their intermittency is essential and must coincide with the observation period (Plougonven et al., 2017). Geolocalization is also a problem often faced when studying the atmosphere; for example, satellites can provide global coverage and seasonal variations of the observations but fail to

study local features and properties, and the wave movement (Ern et al., 2004; Alexander et al., 2015). Also, the computation treatment applied to the data can limit the data; for example, an average generally suppresses wave information on a series.

Airglow imaging is one of the most employed and valuable techniques for observing gravity waves (Swenson and Mende, 1994). Airglow is the atmospheric emission emitted by some constituents excited by photochemical reactions. Many works have used airglow to study gravity waves in the mesopause region and study their general characteristics and parameters throughout

the seasons (Nakamura et al., 1999; Essien et al., 2018) and their general dynamical related to the background conditions (Medeiros et al., 2003; Giongo et al., 2018). To obtain the wave parameters and characteristics, most works employed spectral analysis based on Fourier transforms applied directly on the image to determine the wave parameters (Garcia et al., 1997; Giongo et al., 2020). An airglow camera is relatively low-cost and works for extended periods; aside from its limitation to night-time observations, it can provide information on small-scale waves and long-term features of the mesopause region

(Taylor et al., 2007).

The keogram technique was used with airglow images to study long-period waves (Paulino et al., 2011; Figueiredo et al., 2018). The method allowed the study of the waves above the observation sites for many consecutive years. Those places had many observations of small-scale waves and could extend the observational spectra to the region of the medium-scale waves (Taylor et al., 2009; Bilibio, 2017; Essien et al., 2018).

Observations of medium-scale waves over the Antarctic continent by airglow have not been done until now. Besides, observations done by satellite might comprise waves with similar spatial dimensions but mainly in the stratosphere (Hoffmann et al., 2013; Hindley et al., 2019), or using a limb-viewing technique (Alexander et al., 2009; Hindley et al., 2015), therefore limited to vast horizontal scales. Airglow observations have the advantage of high temporal resolution in a fixed location, allowing the identification of faster and slower waves to the exact spatial dimensions (Bageston et al., 2009; Nielsen et al., 2009; Kam

et al., 2017). In this way, extending the airglow observation to medium-scale waves will allow us to unravel which phase speed spectra those waves have, as done by Matsuda et al. (2017) for small-scale waves.

This work used the keogram technique to study the medium-scale waves observed by airglow imaging over the Antarctic Peninsula. The main focus of this paper is to present the analysis method to ensure the reliability of the analysis of the waves for posterior use at other observation sites and constrain the observations concerning the whole wave spectra for comparison

with different instruments and techniques. Section 2 describes the data used in this work and explains a new methodology to



analyze the medium-scale waves in keograms, including wavelet amplitude correction and phase quality-of-lag identification. Section 3 tests the methodology using synthetic image simulation to estimate errors and validity of results. Section 4 shows the results, and section 5 drives the discussions on the results and simulated methodological features. Section 6 is the conclusion.

## 2 Methodology

### 2.1 Image Processing

The data utilized were airglow ground-based images obtained by an airglow all-sky imager installed at the Brazilian Antarctic Comandante Ferraz Station (CF or EACF) on King George Island, 200 km from the tip of the Antarctic Peninsula. The imager has a 3-filter wheel for OH-NIR (700-900 nm), OI 557.7 nm, and 630.0 nm emissions. The camera is cooled to -80 °C degree and has 80% quantum efficiency for visible and near-infrared wavelengths (KEO SCIENTIFIC LTD, 2022).

For medium-scale wave parameter estimation, the following procedures were applied to the images as preprocessing: (a) Calibration of the images to project them onto the geographical coordinates ; (b) Star removal; (c) Flat-fielding of the images; (d) images subtraction ; (e) Keograms construction ; (f) Butterworth filter to remove noise and avoid long-term oscillations due to the twilight and moon. Each procedure is described in detail below.

The calibration of the images used in this work follows the well-established and widely used procedures developed initially by Garcia et al. (1997) and lately improved by Wrasse et al. (2007) with other minor improvements by other authors such as Bageston et al. (2009); Giongo et al. (2020). The calibration includes aligning the images with the north at the top of the image (east to the right) and projecting them into geographical coordinates on a 1x1 km grid, resulting in 512x512 km images of the airglow layer centered on the station. The star removal is based on a statistical inference of the neighborhood of the peaks to remove them without interfering with the amplitude of the oscillations in the background. Van Rhijn and atmospheric extinction effects are corrected together with flat-fielding to avoid brightness variations due to the observation angle of the all-sky lens fitted to the camera (Kubota et al., 2001; Wrasse et al., 2024). Consecutive images are subtracted in order to suppress the Milk Way and slow varying features in the images (Tang, J. et al., 2005; Vargas et al., 2021).

Figure 1 shows an example of the original image obtained by the imager and the resultant preprocessed images after the above-described procedures. Fig 1a is the original image, 1b is the star-removed image, 1c is the Van Rhijn and atmospheric extinction corrected image, and 1d is the projected image onto a 512x512 km grid. Observational and distortions due to the lens were successfully removed. Milk Way is an inconvenient feature in the images that will not be a problem for the analysis, as discussed later.

The keograms are built by selecting the central row and column of the image for one night and disposing of them as a function of the time, which results in a diagram with the long-period oscillations now revealed with the zonal and meridional components (Taylor et al., 2009; Figueiredo et al., 2018). A Butterworth filter is used on each time series (lines of the keogram) with cutoff periods of less than 20 min and greater than 240 min to focus on the possible wave periods that can be observed in the images. These limits were selected based on previous knowledge of the limitation of the technique: shorter periods representing high-frequency waves easily analyzed by other methods; and longer periods (>4 hours), which might exceed the





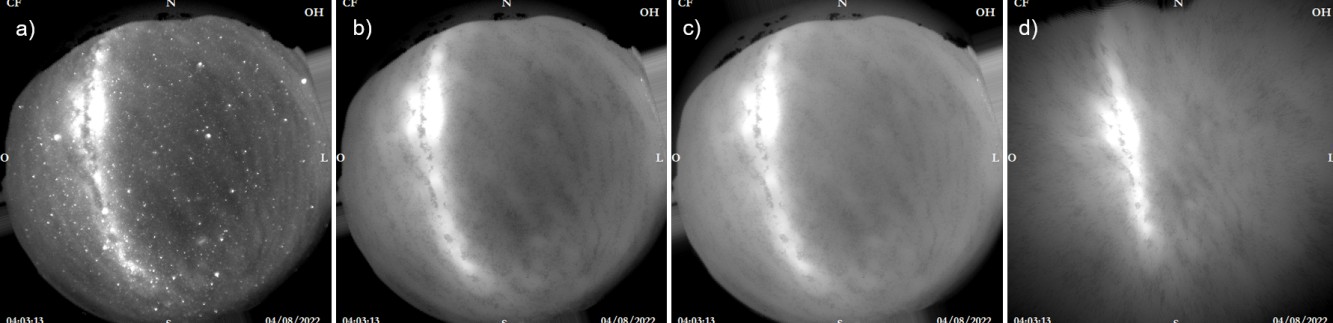

**Figure 1.** Example image obtained with the all-sky camera and the resulting image after the processing described.

duration of observations, are excluded from the analysis. More details about the typical observation length and the cone of
influence (COI) are discussed later.

Figure 2 shows an example of a post-processed keogram, as described above. The top panel is a zonal keogram made by the
horizontal slices of the images, and the bottom panel is a meridional keogram made by vertical slices. The time axis shows the
maximum observation length of the night at EACF, so the blank space is due to the restricted observation time for this specific
night. Oscillatory features are seen in the keograms modulating each other and changing their properties along the night.

## 2.2 Spectral analysis

In this work, we developed a new method to analyze the waves based on the wavelet of Torrence and Compo (1998) to
extract the parameters. Although the procedure is based on Fourier Transform as in previous works, the new feature of this
methodology is that it automatically selects the waves along the keogram and verifies the quality of the oscillatory signal,
resulting in less user bias on its usage. Below, we describe the details of the algorithm involved in the spectral analysis and
temporal and spatial selection of the waves on the keogram simultaneously.

First, a Morlet wavelet transform is applied to the central line of the keograms, which is the series coincident with both
keograms because it is the zenith pixel series. Peaks in the wavelet power spectrum are selected using a peak selection procedure
to identify the significant oscillations and their temporal position. Reconstruction of all time series is performed for every period
determined in the zenith wavelet, resulting in one reconstructed keogram for each prominent period.

Figure 3 shows the wavelet power spectrum for the zenith time series of the keogram of Figure 2. Prominent peaks can
be seen near 20, 30, 50, 80, and 140 minutes of period, with temporal variation and combination among them. The peak
identification procedure needs an equally dimensioned array. A linear spline interpolation is performed in the vertical columns
of the spectrum to give it the same number of points in the vertical as in the horizontal lines.

Spectral phases along the columns are selected to calculate the average phase tilt along the space, which is a consequence
of the movement of the waves in the images. The phase difference is the value used to calculate the wave's wavelength and the
phase speed afterward. The temporal position chosen to calculate such phase difference is around the peak in the zenith wavelet.



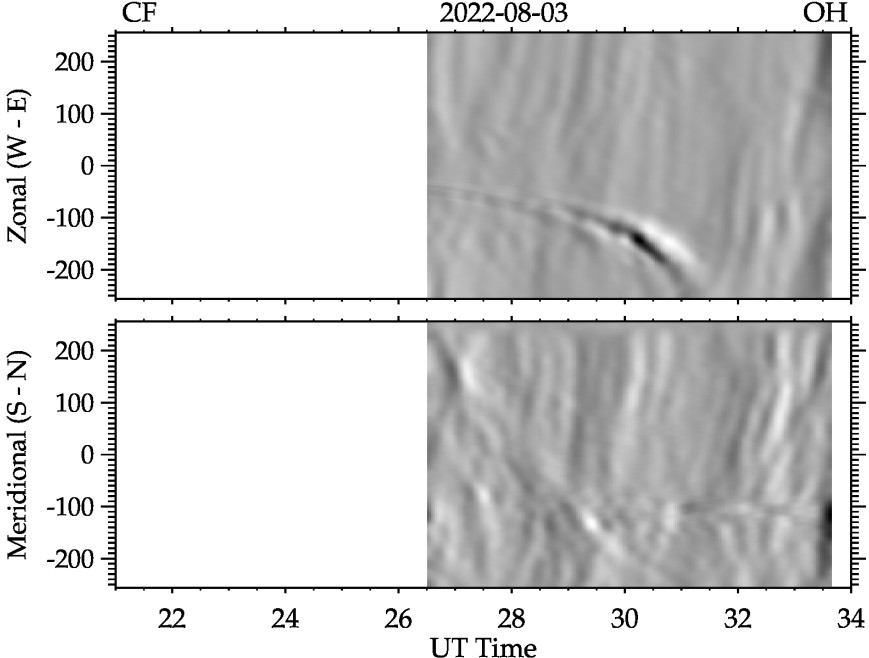

**Figure 2.** Example keogram with all the described pre-processing procedures applied. At the top, the zonal keogram is made with west-east cuts in the images; at the bottom, the meridional keogram is made with north-south cuts.

Several lines are selected half period long before and half period long after the peak central position. A linear fit is applied to the phase lines, in which the angular coefficient is used to calculate the wave parameters as it has the angular difference information, and simultaneously, the fitting has information for checking the signal quality for the selected reconstructed period.

In this way, the fitting information can indirectly determine whether the signal is potentially a coherent wave. A minimization value is set to filter the location in which the fitting is reliable and ensure the signal is due to a traveling wave. Nearby phase lines are used to perform the same fitting procedure and, subject to the same minimization threshold, are used to calculate the wave parameters' average around the spectrum's peak time.

Figure 4 shows the reconstructed keogram for the peak period of 35 min, which appears between 29 and 30 UT. Spatial
and temporal changes in the amplitudes are seen along the keogram. Vertical red lines are the phase columns selected to calculate the linear fittings, and their vertical position is the best calculated fitting position for the central column phase. In both components, the selected region seems to indicate a coherent wave signal.

Figure 5 shows the phase lines around the temporal peak of the 35-minute wave. The phase lines were unwrapped to better visualize the tilt along the columns and avoid misfitting due to the cycle ambiguity. The red sectors highlight the region where
the best fitting was found, and they coincide with the red vertical lines in Figure 4. The best-fitting region criteria automatically selected the phases' most flat and parallel sections.





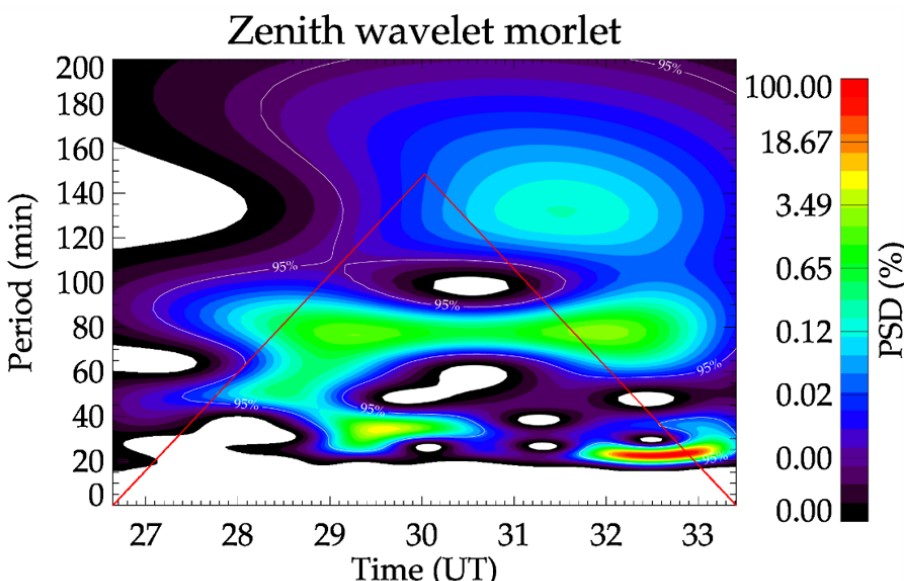

**Figure 3.** Zenith series wavelet power spectrum normalized in percentage.

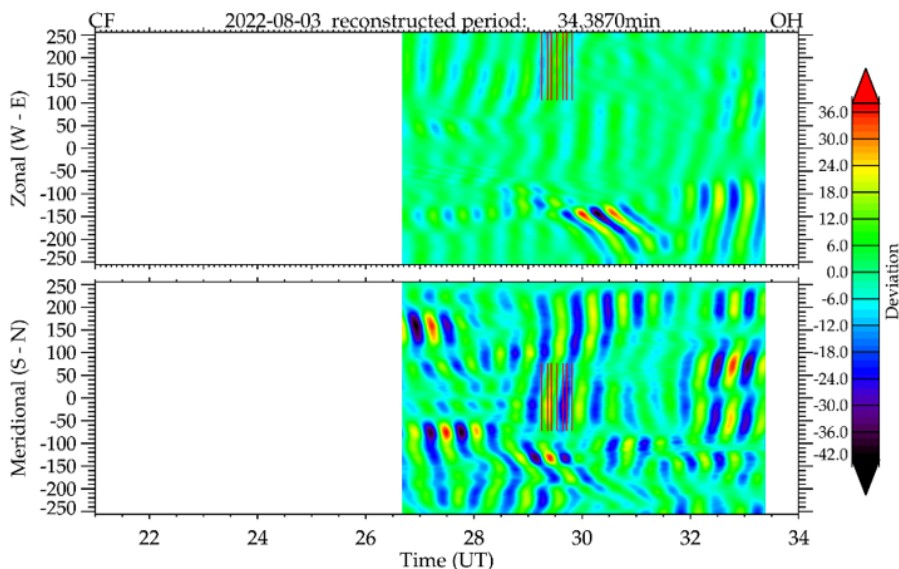

**Figure 4.** Reconstructed keogram for the 35 min period peak, visible between 29 and 30 UT.

After the determination of the correct phase lines and their position, the parameters are estimated as follows: i) the phase difference along the space ($\Delta\psi$), which was determined by the fitting process (it is also the angular coefficient of the fitting), is weighted to calculate the wavelength components by the equation:



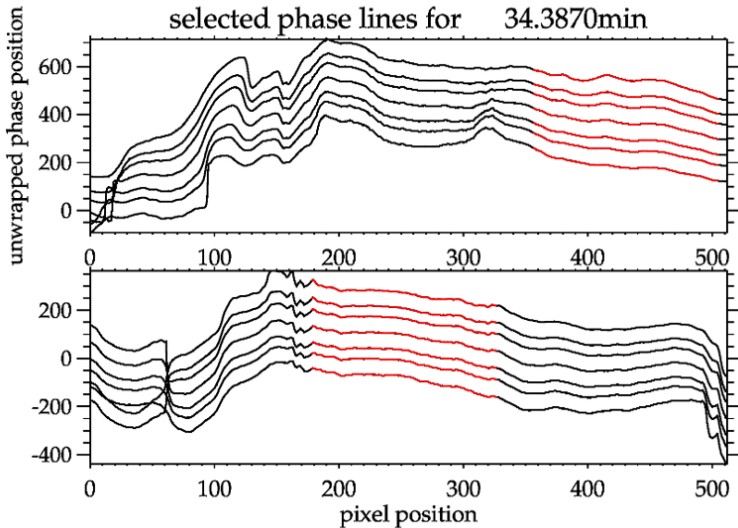

**Figure 5.** Phase lines half period away from the peak position on the time coordinate. Those lines were selected from each reconstruction and unwrapped to avoid cycle ambiguity.

$$\lambda_{NS,EW} = \frac{\Delta d}{\Delta \psi / 360},$$ (1)

where $\lambda_{NS,EW}$ is the wavelength components (EW for the zonal keogram and NS for the meridional one), and $\Delta d$ is the distance between the lines of the keograms (spatial resolution of the images). The horizontal wavelength $\lambda_H$ is then calculated using geometry:

$$\lambda_H = \frac{\lambda_{NS}\lambda_{EW}}{\sqrt{\lambda_{NS}^2 + \lambda_{EW}^2}}.$$ (2)

ii) The period is estimated directly by the peak position and was also used to make the reconstructions;

iii) The phase speed is calculated as a function of the wavelength and the period following the relation:

$$C_H = \frac{\lambda_H}{\tau}.$$ (3)

iv) The geometry of the wavelength components finally estimates the wave phase propagation direction:

$$\phi = COS^{-1}\left[\frac{\lambda_H}{\lambda_{NS}}\right],$$ (4)





where $COS^{-1}$ is the function that retrieves the angle in which its cosine is the input. The results are validated by checking the phase lines' reconstruction quality, flatness, and parity. The purpose of this last step of checking is discussed further in the simulations.

## 2.3 Amplitudes estimations

The wavelet transform can directly estimate the amplitudes of the waves, although it requires corrections due to the filters
applied to the data and wavelet bias, as reported previously (Torrence and Compo, 1998; Liu et al., 2007). To correct the wavelet biasses, we followed the correction performed and studied by Xu et al. (2024, and references therein), which is dependent on the scales (periods) and image resolution given by:

$$C_f = \left[\frac{\delta t}{2\pi^{1/2}s}\right]^{1/2},\tag{5}$$

where $\delta t$ is the temporal resolution, $s$ is the scale (usually within 3% difference from the periods for morlet).
Also, for the time difference images, a correction following the work of Tang, J. et al. (2005) was used, which is dependent on the resolution (time difference between the images) and frequency of the waves present in the images, given by:

$$I_{td}/I_0 = 2sin\left[\frac{\omega\delta t}{2}\right],\tag{6}$$

where $\omega$ is the wave's frequency (inverse of the period). These corrections discussed here are applied after the peak identification.

For the example used in this section, Table 1 presents the results for the night of August 8th. The procedure approved three waves with periods of 34, 79, and 45 minutes. Other peaks in 134 and 22 minutes were identified but did not match the fitting conditions for the phase. Waves 1 and 2 have very small standard deviations on their parameters. Although wave 3 has more dubious results, it still is below 15% of error.

**Table 1.** Results of the example wave shown in the methodology overall.

| Parameters | $\lambda_h$(km) | $\tau$(min) | $c_o$(m/s) | Az (°) |
|---|---|---|---|---|
| Wave 1 | $402 \pm 26$ | $34.9 \pm 1.38$ | $194 \pm 12.63$ | 44 |
| Wave 2 | $529 \pm 6$ | $79 \pm 3.16$ | $111 \pm 1.4$ | 46 |
| Wave 3 | $260 \pm 34.3$ | $45.3 \pm 1.8$ | $95.7 \pm 12.6$ | 350 |

## 3 Synthetic Images Simulations

Error estimations and procedure quality were developed by simulating Synthetic waves in the image. Artificial images made as arrays containing known wave patterns were used to verify the quality of the analysis, track the error propagation through





the procedure, and estimate future errors generated throughout the process. The images are created with the desired wave properties, i.e., wavelength, direction of phase propagation, and period. Also, a wave movement can be applied when making a sequence of images, considering phase speed and time difference between images (time resolution). It is important to emphasize

that this study is not trying to validate the wavelet transform or other well-known mathematical procedures, like linear fitting. Still, it tracks errors throughout the combination of processes employed. Errors due to any transform or mathematical procedure exist and can be studied or found in other publications (Liu et al., 2007; Xu et al., 2024).

   Figure 6 shows an example of a keogram generated from synthetic images. In this case, two waves are superposed with wavelengths of 100 and 250 km, periods of 50 and 70 min, phase speeds of 55 and 100 m/s, and propagation directions of

150° and 60° azimuthal angle. Noticeably, the waves have the same propagation direction in the zonal component, while the opposite direction is in the meridional component. Also, wave amplitude combinations are evident in both components.

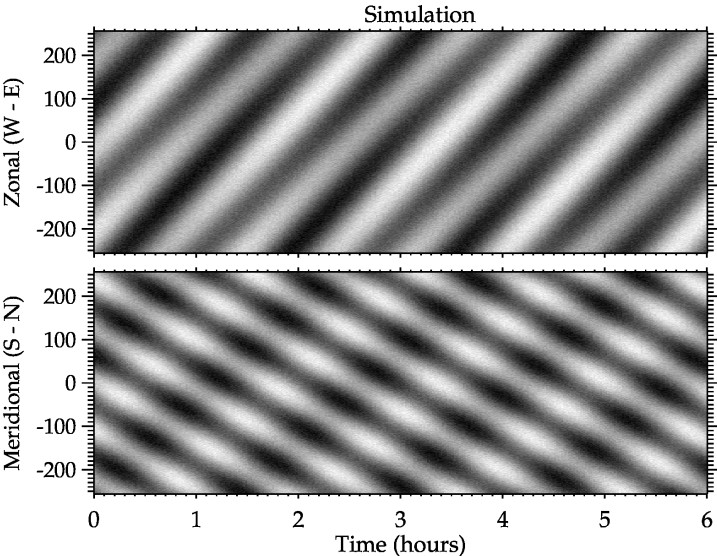

**Figure 6.** Keogram generated from synthetic images with the desired wave properties.

   Figure 7 shows the wavelet power spectrum of the synthetic keogram displayed in Figure 6. Even with the wave uniformly on all the keogram, amplitudes decay out of the cone of influence. The red lozenges indicate the identified peaks in the periods for which the program will test the amplitudes and the parameter estimations.

Figure 8 shows phase lines for the 50-minute simulated wave. Note that the main reason for the phase tilt's nonlinearity is another wave's coherent presence. The lines become irregular as the waves superpose in periods or wavelengths. Amplitudes could also interfere, and a prominent wave can completely occult a smaller one. This discussion followed much of the Fourier transform properties, and our study provided very carefully driven insights into the deviations generated by wavelet phase reconstruction.





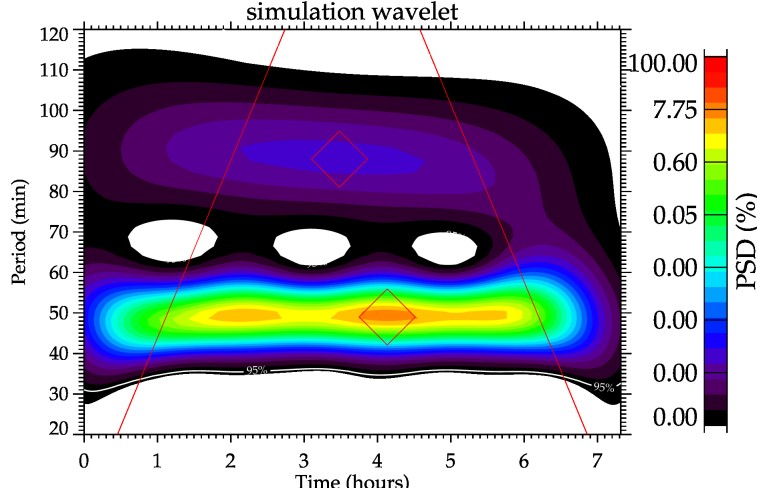

**Figure 7.** Power spectrum of the simulated keogram.

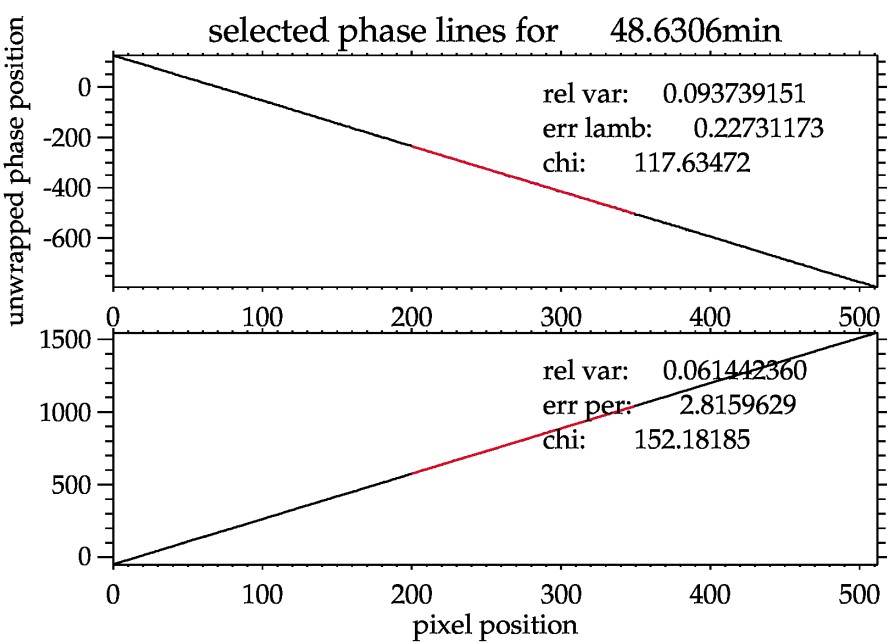

**Figure 8.** Selected phase line for the 50-min wave in the simulated keogram shown in the previous figure.

Figure 9 shows the errors in the calculated parameters as a function of the variance of the angular parameters of the linear fitting. More than 800 keograms simulations were run to make these tests. The red line passes through the 0 point to the 15% in variance equals one, but it seems curved due to the log scale in the x-axis. Numbers inside each plot show the relative



number of waves over 15% of the error to a maximum threshold (in this case, its value equals one). Almost linear growth of the errors with amplitude is seen with the variance increase, except for the periods and the amplitudes that follow a regular

value. Deviations in the periods are due to intrinsic limitations of the wavelet transform, and amplitudes will be discussed later. The main pattern we can stipulate here is a possible cut-off value for the variance to guarantee the correct wave selection and accurate parameter estimation. This threshold could be 1 for the shown case, but its value must be adjusted for each case due to the variance sensibility to previous processing. Conditions that can interfere with the variance include the presence of other waves, noisy data, resolution of the data, and observation duration. Those conditions were included in the simulations and

verified as genuine but are not shown here.

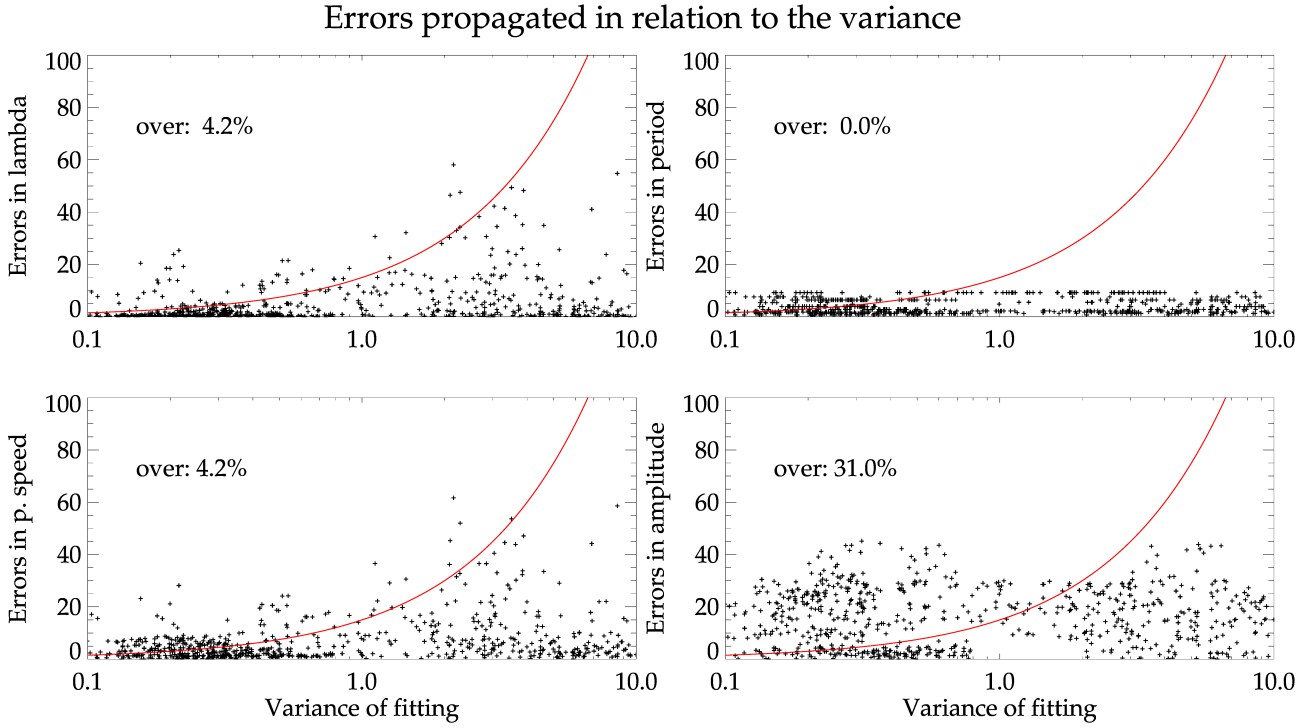

**Figure 9.** Errors in the final result of the simulations as a function of the Variance of the linear fitting applied to the phase lines of the reconstruction.

Amplitude correction is delicate due to the wavelet's numerical limitations, and even when corrected, they still show many problems. Wave superpositions are the main problem, and a substantial decrease is seen when the spectrum's peak lies off the cone of influence. Amplitudes were evaluated separately to recognize why they were much bigger than the other parameters. As the amplitudes do not need to pass through the phase line identification, their errors were evaluated with the distance to the

COI. The conclusions were that peaks inside the COI have less than a 10% error in their estimation, while outside of the COI can reach 20 %, with no dependency on the distance of the COI. This addresses the problem of the peaks outside the COI, in





which no significant errors could be found for selecting outside the COI, except for the amplitudes that grow errors to about 20%.

Using the properties shown in Figure 9, the errors can be inferred by taking a mean value of the errors provided by the simulations, considering the threshold equals one. Table 2 shows the average values retrieved by the simulations and limited by the discussed threshold. Except for the previously discussed amplitudes, those error values do not grow much ahead of the threshold.

**Table 2.** Average results on the error propagation study conducted by simulations.

| Parameters | $\lambda$ | $\tau$ | $C_o$ | Amplitude |
|---|---|---|---|---|
| Errors (%) | 2.56 | 3.61 | 4.24 | 15.28 |
| Standandart deviation (%) | 4.31 | 2.59 | 4.55 | 11.67 |

## 4 Results

In 2022, an airglow camera operated, capturing images on nights when the moon was absent from the observable sky. To analyze medium-scale waves and have reliable results, only observation windows longer than 2 hours were used to ensure the observed waves lie close to the cone of influence and ensure no bias in the amplitude estimation. Following this idea, 18 valuable windows were selected for analysis, which led to 46 medium-scale waves analyzed with the described methodology.

Histograms of the obtained parameters are displayed in Figure 10. The histogram of the observed horizontal wavelengths is shown in the first panel, while the second and third panels show the histograms of the observed periods and phase speeds. The last panel shows the relative amplitudes of the waves, that is, the amplitudes derived by the wavelet analysis divided by the mean background brightness of the images. Besides a large spectrum of wavelengths from 100 to 1000 km, most waves had less than 400 km wavelengths. Periods concentrated at a peak of 20 to 90 min (20 min was the lesser value accepted in the analysis), with some cases going to almost 180 min. Phase speeds were concentrated around a 75 to 200 m/s peak, with some slower waves. Amplitudes were smaller than 4% with concentration below 2%; those waves are very faint oscillations.

Figure 11 shows the propagation directions of the waves as a function of their respective phase speeds, that is, their phase velocity vectors. Waves for a broad spectrum of velocity propagate in all directions without apparent anisotropy. However, more waves are seen with zonal components preferentially in the eastward direction compared to the westward direction. In addition, no filtering pattern could be identified.

## 5 Discussions

Two central problems are observed in the real keograms that are not present in the simulated ones: 1) the waves are usually not everywhere in the keogram; 2) there is much interference with noises and other waves. Superposition of the waves is inevitable, as is an intrinsic problem in the oscillator theory; if some of the wave properties are too close, like period, wavelength, or direc-



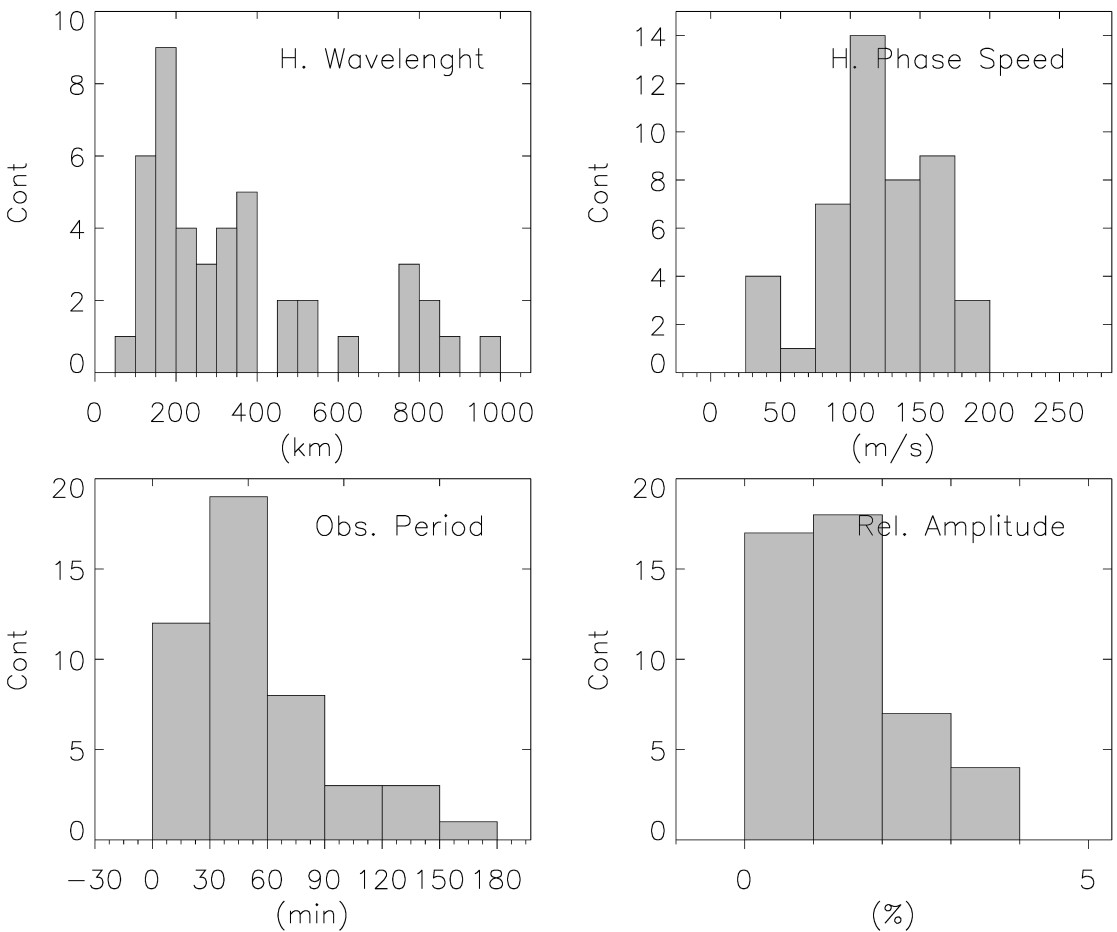

**Figure 10.** Histograms of the observed parameters over the global number of waves.

tion, the final result can be a combination between the two waves, as was verified in the simulations. Although such problems can not be solved easily in an automatic procedure, we could avoid them by visually checking the reconstruction to confirm the presence and coherence of the waves. Also, the threshold used in the phase lines fitting deviation helped automatically avoid noisy and combined waves.

In Figure 9, many higher values of errors are present all over the spectrum due to the misselection of the wave properties that become ambiguous when waves interfere. The resulting period can be closer to one wave while the resulting wavelength is closer to another. The analysis identified the period of one wave but the wavelength of another. When this happens, the interference makes the phase lines visually tortuous. Such conclusions led us to believe that a correct selection of the waves



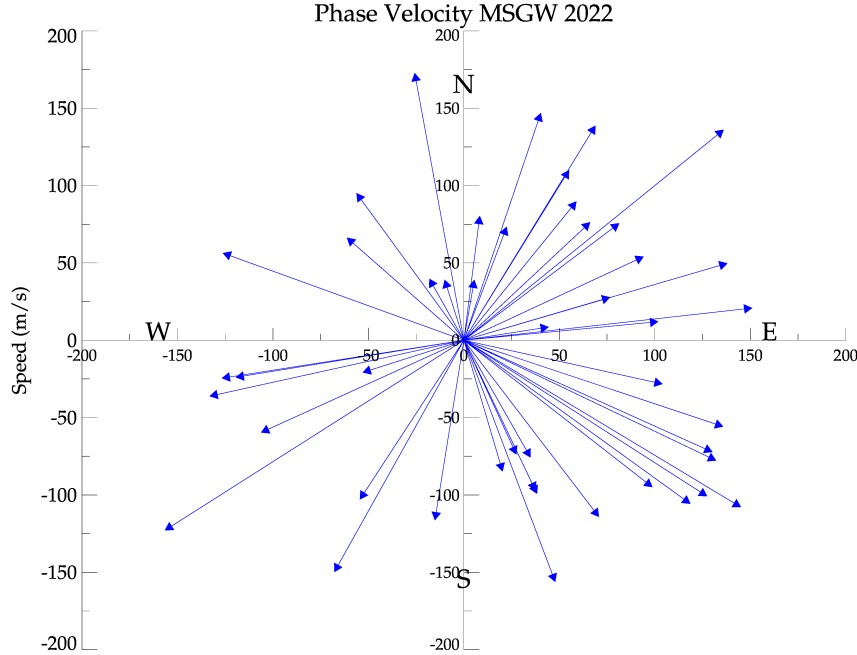

**Figure 11.** Phase velocity vectors of the analyzed waves. Each blue arrow is one wave phase velocity vector.

must follow the check of the reconstruction and the absence of tortuous signal in the phase lines (they can be bent but not tortuous, which is a consequence of a close superposition of the waves).

Finally, we assume synthetic images propagate the same errors as real images when considering their discrete signal. In conclusion, the error estimation determined in the simulations is used as the inherent error propagated in the analysis and will be used for future works in the real keograms. Errors in the phase speed are naturally more significant due to the error propagation. About the peaks outside the COI, no significant errors in the parameters could be found for selecting the peak outside the COI, except for the amplitudes that have regular errors around 20%. Although those errors are significant, the overall methodology is very satisfying and delivers impressive results comparable to the previous methods. It could also reduce user bias by automating most of the process.

In the tropical Brazilian central region, medium-scale waves were observed by Taylor et al. (2009) by airglow imaging using a different methodology. Their results showed waves with horizontal wavelengths of 50 to 200 km, 20 to 60 min observed periods, and observed horizontal phase speeds from 20 to 80 m/s, smaller values than ours and more concentrated. Most of their waves were going southeastward and eastward. Above the equatorial region Essien et al. (2018) observed an extensive amount of waves using a Fourier Transform to analyze the waves. The waves above the equator had 50 to 250 km of wavelength,



periods of 20 to 80 min, and speeds of 20 to 120 m/s, with a majority of the events going north and northeast; they also could observe a strong anisotropy in the propagation directions related to wind filtering. Gravity waves observed in this work differ from other observation locations, mainly in the phase speed and the distribution of the wavelengths. The phase speed showed a more extensive distribution with larger values. At the same time, the wavelengths had a more extensive peak distribution, including a secondary peak of around 800 km, which was not yet reported to the authors' knowledge in airglow observations.

Amplitudes of the medium-scale waves observed by airglow imagers have not been addressed until now. Vargas et al. (2021) studied waves with periods similar to ours directly in airglow images and estimated the amplitudes of the waves with less than 50 min having generally less than 5% of relative amplitudes; that is, their results match with what we observed. The medium-scale waves have small amplitudes, which seems reliable since they are dainty oscillations on the background of the small-scale waves. The study could identify those faint waves through the applied image processing and distinguish superposing waves by

the phase line identification.

## 6  Conclusions

A new analysis methodology was developed for medium-scale gravity wave observations done by keograms of airglow images. The procedure is based on wavelet transform and uses its properties to extract the wave parameters without significant problems or error propagation. Wavelet amplitudes were corrected to certify reliable wave amplitude estimations.

Synthetic image simulations were used to verify the error propagation along the procedure and ensure a reliable parameter estimation for the waves in the range of medium-scale gravity waves. Error propagation is small for all parameters except for the amplitudes, which, besides their correction, can still be underestimated when the peak relies outside of the cone of influence. Still, this behavior does not relate to the distance to the COI.

The results showed the typical behavior of the wave parameters observed in other observation regions using different analysis

methodologies. In a general sense, the analysis methodology showed incredible performance in identifying the waves and estimating the parameters of the waves, also reducing the user bias, which was generally highly user-oriented in previous methodologies. The flaw of the method is a threshold dependence in the fitting quality estimation, which is also dependent on image processing and resolution of the data but was shown not to be a limitation of the method nor the quality of the results. Besides its flaws, the new methodology is ready to be applied to other observatories along the Antarctic continent, a planned

activity currently being completed.

*Data availability.* Airglow images of the Comandante Ferraz Antarctic Station are available at the EMBRACE/INPE website.



*Author contributions.* Conceptualization, GAG, JVB and CMW; Data curation, JVB, CMW, PKN, CAOBF, AB, and DG; Formal analysis, GAG and PDP; Methodology, GAG, CMW, CAOBF and AB; Software, GAG, CMW, CAOBF; Validation, CMW and PDP; Visualization, GAG, CMW, PDP and HT; Writing – original draft, GAG; Writing – review & editing, GAG, CMW, PDP and PKN.

*Competing interests.* The authors declare that they have no conflict of interest.

*Acknowledgements.* This work is part of the First Author's Ph.D. studies at the National Institute for Space Research (INPE), supported by CNPq (140401/2021-0) and Coordination for the Improvement of Higher Education Personnel (CAPES) finance code 001. The authors acknowledge the FAPESP Project under the grant 2019/05455-2 and the support of the Brazilian Antarctic Program (PROANTAR), which has guaranteed the re-establishment of the Airglow observations at Ferraz in 2022. G. Giongo thanks the CAPES for the exchange scholarship, 280 financial code 001, and CNPq for grant number 140401/2021-0. PDP would like to acknowledge the National Science Foundation grant 1443730. The authors acknowledge the Brazilian Study and Monitoring of Space Weather (Embrace) Program at INPE for providing the all-sky airglow images used in this work.



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
