# Peer review of "A new approach to characterize the medium-scale gravity waves using Antarctic airglow observations"

_EGUsphere, 2024_

## Referee Comment (RC1)

The paper by Giongo et al. investigates the activity of medium scale gravity waves over the Antarctic Peninsula by developing an algorithm that identifies the waves automatically on the airglow images and estimates their parameters. This automatic process is based on the well known keogram analysis. Although new approaches to improve image analysis are welcome, there are issues that prevent the paper being published in the present form.

Although the paper presents some results on medium-scale gravity waves, its main focus is on the method used to identify the waves and estimate their parameters. However, the description of the method lacks clarity, and the results obtained are not adequately discussed.

As the authors propose a new method to analyze images and extract wave parameters, they should be as clear as possible in explaining it. The description of the spectral analysis procedure lacks a clear and logical sequence, making it difficult to follow. While the authors claim the process is fully automatic, it appears to require user intervention at certain stages. To improve clarity, the steps of the analysis should be explicitly defined, distinguishing between automated and manual actions. Ensure that the steps of the analysis are presented in a logical order, from data acquisition to final output. If the process is automatic, specify which steps are handled by the system and which (if any) require user intervention. Additionally, the method should be tested by calculating the wave parameters as in standard keogram analysis, as quoted in the manuscript, and comparing the results with those obtained from their automated method.

**Specific questions**.

At page 4, lines 101-103, the authors state that "Peaks in the wavelet power spectrum are selected using a peak selection procedure to identify the significant oscillations and their temporal position". Please describe the details of the procedure.

Is the central line of the keograms the horizontal line in Figure 2 that intersects the vertical axis at the zero mark?

In Figure 2, does the vertical axis represent distances from the central point of the image? If so, it should be labeled with appropriate units.

The following sentence needs some clarification.
 "**The peak identification procedure needs an equally dimensioned array. A linear spline interpolation is performed in the vertical columns of the spectrum to give it the same number of points in the vertical as in the horizontal lines**".
What are vertical and horizontal lines?
I understand that wavelet analysis demands time series equally spaced. The authors inform that they use interpolation. Are the images equally sampled? i.e. is the interval between two images always constant?
What is the source of the difference between the number of vertical and horizontal data points?

Each data point of the time series is the pixel intensity along the night. What is the source of the unequal time resolution of the zonal and meridional keograms?

What the authors mean in the phrase "**with temporal variation and combination among them**", in the sentence "**Prominent peaks can be seen near 20, 30, 50, 80, and 140 minutes of period, with temporal variation and combination among them**", at page 4, lines 105-106?

The following sentences are unclear. "The peak identification procedure needs an equally dimensioned array. A linear spline interpolation is performed in the vertical columns of the spectrum to give it the same number of points in the vertical as in the horizontal lines".

Do the vertical columns correspond to the meridional keogram and the horizontal ones to the zonal keogram?"

On page 5, line 112. Does the sentence, "**Several lines are selected half period long before and half period long after the peak central position**",  mean the authors selected a symmetrical time window centered on the peak position, extending half a period before and half a period after the peak? From which figure are these lines selected?

It is not clear how the keogram in figure 4 is reconstructed. The wavelet analysis takes into account only the central line (which is the same for zonal and meridional directions) of the keogram. How are the keograms off the central line reconstructed?

In Figure 4, there are vertical red lines indicating the selected lines. Are these lines inserted manually?  Those lines are placed at distinct spots in the zonal and meridional keograms. How is their placement?

In Figure 5, what is the meaning of the red phase lines? The figure legend does not inform it. Additionally, the vertical axis lacks units, which I suppose should be kilometers.

Regarding the two synthetic waves used to test the method. Do they have the same amplitude? The wavelet spectrum in Figure 7 shows distinct power for the waves.

Figure 8, what are the units of the vertical axis?

Authors state that they used "**more than 800 keograms to run the tests of method**". How were those keograms generated? Do the synthetic waves have the same properties (e.g. wavelengths, periods, amplitudes)? Explain the distribution of the wave parameters.

At page 11, lines 191-196, the authors state some issues related to the amplitude of the synthetic waves retrieved through their method. The main reason is the superposition of the waves. In the atmosphere, waves do coexist, making this an issue that is difficult to avoid.

On page 12, lines 215-218, the authors suggest that the direction of propagation is isotropic, but say that the waves propagate preferentially eastward. It seems contradictory. Additionally, state that there is no filtering pattern. I infer they are referring to filtering by background wind. It is important to make this clear. Moreover, explain why such filtering is not observed for these waves.

On page 14, line 240-244. The expression "**Above the equatorial region**" is used a couple of times referring to observations at Cariri (7.4° S). This phrasing sounds unusual.

On page 15, lines 250-255. Amplitudes of gravity waves inferred from airglow are rare because the all-sky imagers must be calibrated to measure absolute intensity.

The authors analyzed gravity waves observed in the Antarctic Peninsula and compared them with waves observed in the equatorial region. For the purpose of the paper, which I assume is to validate their methodology, this comparison may not be ideal.

The keogram technique is widely used to investigate medium-scale gravity waves in airglow images. Since the authors are proposing an automatic algorithm, they should validate their results against the standard keogram analysis by manually analyzing the wave parameters.

**Minor**
page 3, line 63
63ºC degree -> 63ºC
page 3, line 79
Fig 1a -> Figure 1a
page 3, line 87
disposing of them -> disposing them
Maybe the word "disposing" is not the best choice to describe the situation. It could be "arranging them", "stacking them".
page 3, lines 87-88
shorter periods representing high-frequency waves easily analyzed by other methods -> shorter periods representing high-frequency waves can be easily analyzed by other methods

page 4, lines 105-106
Prominent peaks can be seen near 20, 30, 50, 80, and 140 minutes of period ->Prominent peaks can be seen at periods near 20, 30, 50, 80, and 140 minutes

page, 5, line 118 average around ->averaged around

page 12, line 212, 20 min was the lesser value ->20 min was the lowest value

---

## Referee Comment (RC2)

**Review of: "Medium-scale gravity waves observational methodology for antarctic airglow observations" by Gabriel Augusto Giongo et al.**

**Overview**

This manuscript introduces an improved method for detecting medium-scale gravity waves from ground-based airglow imagers. The method is based on keogram analysis, it is designed to automatically detect wave events and determine the horizontal wavelength, azimuth, period and ground-based phase velocity of the waves. It was tested using simulated data as well as actual airglow images. Authors report reliable wave identification and low errors of the retrieved wave parameters.

Unfortunately, the new methods, although they seem to contain good ideas for dealing with many aspects of the problem, are not explained clearly enough, to the point that I had difficulties interpreting some of the main results (see major comments below). I have also outlined some possible issues with the method itself and made a few suggestions, I hope they can be of some use. Since introduction of the improved methodology is the main goal of this paper, I cannot recommend it for publication in its current state.

**General/major comments**

1. **Severe lack of clarity in mathematical methodology and result presentation.** This manuscript cites quite a few previous works in relation to various methods used in wave analysis and error estimation. While it is the accepted norm for *derivations* of those results and their finer detail, the authors of this manuscript rely solely on the references and very general methodological concepts (e.g. "error propagation") to describe the key quantitative parameters and mathematical methods used in their work. Because of this, I had difficulty understanding which physical or statistical quantities are described by the provided numerical values. Some of the most important results are presented in figures with undefined color scales, or as a percentage of an unstated quantity or as a standard deviation, without a clear a description of the data set for which the standard deviation was calculated (see major comment 2 and specific comments for concrete examples). I must apologize if I overlooked something or if some of the subsequent comments are simply a result of me interpreting some physical quantity not in the way that the authors intended, but, in my opinion, reader should not need to guess quite so often.

2. **Simulated keogram analysis results.** The authors mention "error propagation" in several places in the manuscript. For example, they state that "It is important to emphasize that this study is not trying to validate the wavelet transform or other well-known mathematical procedures, like linear fitting. Still, it tracks errors throughout the combination of processes employed. Errors due to any transform or mathematical procedure exist and can be studied or found in other publications (Liu et al., 2007; Xu et al., 2024)." This leads me to believe that quantities described as "propagated errors" are simply a result of statistical errors of well-know data analysis procedures (e.g. linear fitting, wavelet analysis) combined using the standard methods of calculating the error of a derived quantity from the errors of the parameters the quantity was derived from. On the other hand, running the analysis on simulated data provides an *independent* method of estimating the error: one can simply compare the results of analysis to the known parameters of the simulated data. Therefore:

   (a) I do not understand what authors mean in the beginning of Section 3, when they state that "Artificial images made as arrays containing known wave patterns were used to verify the quality of the analysis, track the error propagation through the procedure, and estimate future errors generated throughout the process". Does this mean that the error estimates for

real data run (i.e. "propagated error") was somehow based on the results of the simulated data run? If yes, then how? Was the simulated data used only for setting the threshold for acceptable linear phase fits, or for something else?

(b) Authors only provide one example of a simulated keogram, while "more than 800" were used in total. More information on this keogram set should be provided. Did every simulated keogram have two waves in it? What were the wavelengths and periods? What were the relative amplitudes?

(c) **(Important!)** Table 2 is extremely confusing. Firstly, I have no idea what the "standard deviation" means here. I did not find an explanation anywhere in the main text. Variance of the linear fit for phase is discussed in detail in this section, but it is not clear how it would be used to obtain the standard deviations for all the measured parameters. Are these just the standard deviations of the parameters used for all the different simulated waves? Secondly, what kind of error estimates are given in the table? Since this is a test on simulated data, it is vital to provide a clear overview of how wave analysis results compare to the actual parameters of simulated waves. Yet the table suggests that this is "propagated error". If this is just the propagated error in the sense I defined it at the beginning of this comment, what was the point of the simulated data run and why are the authors not showing how the analysis results compare to the actual wave parameters? If Table 2 does indeed show this comparison, then how do these errors compare to the errors provided for the real keograms (which are the "propagated errors", right?), where the wave parameters were not known in advance?

3. **Collocation of zonal and meridional wavelength estimates.**

(a) As one can see from figures 4 and 5, the wavelengths can end up being retrieved from a part of keogram that does not include the center (zenith) pixel. This can lead to a situation when zonal and meridional wavelengths are retrieved from completely disjoint parts of the image, neither of which contain the zenith pixel (which is used to identify temporal peaks in wave activity). In my opinion, this a major flaw of the method. Many structures seen, for example, in Figure 4, do not span the whole spatial extent of the keogram: there are major differences in phase structure and phase tilt between North and South, as well as East and West, parts of the image. Therefore, I would assume that method would often estimate the meridional and zonal wavelengths of completely different wave packets. The authors also seem to be aware of this: they state in Section 5 that waves not spanning entire keogram are a problem and that "The resulting period can be closer to one wave while the resulting wavelength is closer to another". It therefore hard for me to understand why so little care is taken to ensure the collocation of both wavelength estimations and the wave activity peak identification. Point (b) outlines one possible way to resolve that, just as an example.

(b) As far as I understand, the method described in the manuscript only makes use of one row and one column from each image. Furthermore, the peaks of wave activity are only identified from a time series obtained from a single pixel. It seems to me, that a simple way to improve the method would be to pick a few (let us denote their number by $n$) equally spaced rows and the same number equally spaced columns in the image. Then one would apply the wavelet power spectrum analysis not just for the zenith pixel, but for the $n^2$ intersections of rows and columns. The strongest peaks in these power spectra could then be used to identify wave events, and wavelengths could be estimated from the meridional and zonal keograms that intersect at the point with a strong power peak. This way, wave events that happen off-zenith could be identified. Even more importantly, one could apply stricter colocation criteria for phase fitting (e.g. only use the parts of the two keograms that contain their intersection point, where the power spectrum peak was identified) without loosing too

many wave events. Finally, if this method ends up detecting waves in close spatial and temporal proximity to one another, one could use this as a consistency check. As far as I understand, none of the methods described in this manuscript are very computationally expensive, so it should not be a problem to run this for $n = 5$, for example. There could be, of course, many more (and probably better) ways to resolve the colocation problems and make better use of data from the whole image.

4. **The need for manual intervention.** In the beginning of Section 2.2 the authors state "Although the procedure is based on Fourier Transform as in previous works, the new feature of this methodology is that it automatically selects the waves along the keogram and verifies the quality of the oscillatory signal, resulting in less user bias on its usage". Therefore, the automatic nature of the new analysis method is the key novel aspect in this work. However, Section 5 implies that some of the most important flaws of the method could be adressed by manual inspection of results. In particular, phase lines of valid wave fits "can be bent but not tortuous, which is a consequence of a close superposition of the waves.". In my understanding, distinguishing a "tortuous" phase line from a bent one should not be too difficult to accomplish automatically: one could consider, for example, checking whether some norm (e.g. the Euclidean norm) of the second temporal derivative of the phase line exceeds a threshold. That would, of course, introduce another ad-hoc parameter to the analysis (besides the currently used threshold for the variance of linear fit), but this would still be more objective than manual inspection of results, that the authors have set out to avoid. Also, was any manual selection of wave fits used to obtain any of the results shown in the manuscript?

5. **Validation against whole images.** Table 2 shows the parameters of some retrieved waves. Those waves have wavelengths that are generally similar, or smaller, than the spatial extent of the airglow image ($512 \times 512$ km). Would it not be possible to verify some of the image parameters obtained from keograms against the corresponding images? I understand that some of the images would be affected by the Milky Way and various other light sources, but would it not be possible to obtain at least some images, where the medium-scale waves discussed would be visible (perhaps after applying low-pass filter to suppress small scale structure)? I agree that keogram analysis has many advantages over this approach, but could individual images be useful at least for demonstrating that wave azimuth and wavelength were retrieved correctly? If this is indeed very difficult, maybe authors could have validated their new method against previous, more manual keogram-based wave analysis techniques?

**Minor/specific comments**

1. L29: I do not understand this sentence. The geographic location and altitude of a particular satellite observation (which is typically meant by the term "geolocation") can typically be determined pretty accurately. In particular, nadir-viewing satellite instruments (e.g. AIRS, AWE missions), offer similar geolocation accuracy as ground-based airglow imagers. Maybe the authors are talking about the spatial data coverage, and not geolocation here? Also, "local features and properties" is an extremely vague description given that methodology is the main topic of this paper! Authors should explain this statement in a lot more detail or remove it.

2. L106: "The peak identification procedure needs an equally dimensioned array". Why is that? Is that a fundamental requirement of the method, or is this just needed for compatibility with some particular implementation, software library, etc.?

3. Figure 2: The color scale in this figure is not defined or explained anywhere. This must be fixed.

4. Figure 3: Percentage of what? Power spectrum is normalized with respect to what exactly?

5. Figure 4: Units for deviation are not given. Also, the choice of color scale is very unfortunate, as there is very little contrast for deviations between 0 and 15, hence only the negative phases of most waves are visible.

6. Equation (5): while it is, of course, OK to refer the reader to the study by Xu for *derivation* of this equation, authors should explain clearly what $C_f$ is and how it was used in this work.

7. L216: "Waves for a broad spectrum of velocity propagate in all directions without apparent anisotropy. However, more waves are seen with zonal components preferentially in the eastward direction compared to the westward direction". In my understanding, the second sentence directly contradicts the first: more waves with zonal components in the eastward direction is anisotropy.

**Minor typos and suggestions**

This is a list of typos that I noticed and minor, mostly language related, suggestions. Point-by-point replies to these are not necessary.

1. L11: I would suggest "The observed parameters of the waves [...]"

2. L15: I would suggest to replace "due to their energy and momentum driving" with "because they transfer energy and momentum", or merge the sentence with the following one, since I find "driving" a bit too vague in this context.

3. L23: "[...] short- and long-term variations of the waves' characteristics as well as short-term variations [...]". The repetition of "short" appears to be a typo.

4. L30: "Also, the computation treatment applied to the data can limit the data [...]". I might have somehow missed the connection with the rest of the paragraph, but, as it is, this sentence is both rather vague and rather obvious. I would suggest to remove it.

5. L67: (d) image subtraction; (e) keogram construction

6. L76: the Milky Way

7. L80: A word may be missing here, as the adjective "Observational" seems to lack a corresponding noun and does not fit into the rest of the sentence.

8. L81: The Milky Way

9. L83: What does "disposing" mean here? Consider rephrasing.

10. L123: The exact meaning of the phrase "lines were unwrapped" was not clear to me.

11. The term "angular coefficient" is not very standard, I had to guess what was meant by that.

12. Section 2.3 title: I would suggest "Amplitude estimations"

13. L157: How are these standard deviations calculated?

14. Section 3 title: I would suggest to write "Synthetic image simulations" or "Simulations with

synthetic images".

15. L181: "The red line passes through the 0 point to the 15% in variance equals one, [...]". I cannot understand what the authors mean here.

---

## Author Comment (AC1)

Authors' response to Referee 1:
Gabriel Giongo et al.

We thank Reviewer 2 for their comments on our manuscript. We acknowledge the significant concerns raised and have undertaken a comprehensive revision to address these issues. Below, we respond to each of the major comments.
Reviewer comments in black, author's comments in blue.

The paper by Giongo et al. investigates the activity of medium scale gravity waves over the Antarctic Peninsula by developing an algorithm that identifies the waves automatically on the airglow images and estimates their parameters. This automatic process is based on the well known keogram analysis. Although new approaches to improve image analysis are welcome, there are issues that prevent the paper being published in the present form.

Although the paper presents some results on medium-scale gravity waves, its main focus is on the method used to identify the waves and estimate their parameters. However, the description of the method lacks clarity, and the results obtained are not adequately discussed.

Considering the general comments and specific questions from both reviewers, we have written a completely new version of the text, which includes a section dedicated to explaining the main steps of the procedure along with the mathematical and physical concepts underlying the transform and its properties. Additionally, the discussions focus on the new features and advancements of the methodology rather than comparisons with other observation sites. The figures have been updated, and a new title has been suggested.

As the authors propose a new method to analyze images and extract wave parameters, they should be as clear as possible in explaining it. The description of the spectral analysis procedure lacks a clear and logical sequence, making it difficult to follow. While the authors claim the process is fully automatic, it appears to require user intervention at certain stages. To improve clarity, the steps of the analysis should be explicitly defined, distinguishing between automated and manual actions. Ensure that the steps of the analysis are presented in a logical order, from data acquisition to final output. If the process is automatic, specify which steps are handled by the system and which (if any) require user intervention. Additionally, the method should be tested by calculating the wave parameters as in standard keogram analysis, as quoted in the manuscript, and comparing the results with those obtained from their automated method.

In the new version, a subsection was added to organize the ideas behind the analysis and provide an overview of the sequence and physical meaning of the steps. The procedure requires no human intervention, but someone can approve the result after it concludes. A flowchart was included in the text to clarify the logical sequence of the procedure for the reader. Discussions now include comparisons with previous methods and advancements in obtaining phase fitting as a parameter for analysis quality, which was lacking in earlier methods.

**Specific questions.**
At page 4, lines 101-103, the authors state that "Peaks in the wavelet power spectrum are selected using a peak selection procedure to identify the significant oscillations and their temporal position". Please describe the details of the procedure.

The procedure has been utilized for over 20 years in numerous publications from our research group, so we did not realize its importance in describing it. However, we agree to include a description of how it works. Details of the procedure are in the text (lines 181-191).

Is the central line of the keograms the horizontal line in Figure 2 that intersects the vertical axis at the zero mark?

Yes, it is also the central pixel of the images over time. The figure was updated to include this information.

In Figure 2, does the vertical axis represent distances from the central point of the image? If so, it should be labeled with appropriate units.

The figure was updated for clarity.

The following sentence needs some clarification.
"The peak identification procedure needs an equally dimensioned array. A linear spline interpolation is performed in the vertical columns of the spectrum to give it the same number of points in the vertical as in the horizontal lines".
What are vertical and horizontal lines?
I understand that wavelet analysis demands time series equally spaced. The authors inform that they use interpolation. Are the images equally sampled? i.e. is the interval between two images always constant?
What is the source of the difference between the number of vertical and horizontal data points?

The peak finding procedure is applied to the Power Spectrum (Figure 3 of the old version, or Figures 5 and 7 of the new one). The horizontal axis of the power spectrum is time, with the same resolution as the images, and equally sampled. The Vertical axis is period, and it depends on the scale (periods) separation of the Wavelet transform, a log-scale-spaced array. So, the vertical is interpolated to the same number of points and equally spaced as the horizontal (time) axis is. Details of the procedure were included in the text (lines 181-191).

Each data point of the time series is the pixel intensity along the night. What is the source of the unequal time resolution of the zonal and meridional keograms?

There is no unequal time resolution between the zonal and meridional keograms; each column of both is the sample row and column of the same square image.

What the authors mean in the phrase "with temporal variation and combination among them", in the sentence "Prominent peaks can be seen near 20, 30, 50, 80, and 140 minutes of period, with temporal variation and combination among them", at page 4, lines 105-106?

The peaks vary in time, which is due to wave interference. The Peak of ~70 min seems to vanish when the 35 and 130 min waves enlarge. For clarity, more discussion on wave interference and analysis reliability was inserted into the text (lines 266-273).

The following sentences are unclear. "The peak identification procedure needs an equally dimensioned array. A linear spline interpolation is performed in the vertical columns of the spectrum to give it the same number of points in the vertical as in the horizontal lines".

Details on the peak identification procedure and the interpolation applied to the power spectrum were included in the text (lines 181-191).

Do the vertical columns correspond to the meridional keogram and the horizontal ones to the zonal keogram?"

In the keogram figure, both components represent in the vertical axis the slices taken along the image, the difference is that the meridional takes a slice in the vertical of the image, and the zonal takes a slice in the horizontal of the image (but it is arranged in vertical on the keogram figure). The horizontal axis is the time at which the image's samples were taken.

On page 5, line 112. Does the sentence, "Several lines are selected half period long before and half period long after the peak central position", mean the authors selected a symmetrical time window centered on the peak position, extending half a period before and half a period after the peak? From which figure are these lines selected?

Exactly. These lines represent phase lines in space over a specific time period. Since the phase spectrum of the wavelet is a function of the same variables as the power spectrum, the phase lines align temporally with the peak (and half period…) and correspond to the vertical lines of the keogram. In other words, a phase spectrum can be associated with each time and space position of the keogram for a given period. The new version of the text describes how the phase spectrum is calculated and how the phase lines are determined and used for the estimation of gravity wave parameters (Section 2.2.1).

It is not clear how the keogram in figure 4 is reconstructed. The wavelet analysis takes into account only the central line (which is the same for zonal and meridional directions) of the keogram. How are the keograms off the central line reconstructed?

They are reconstructed by applying a wavelet transform to them and an inverse wavelet transform only to the selected period set as non-zero, similar to how the zenith is reconstructed. For the period identified in the zenith wavelet, all the lines will be "enforcedly" reconstructed to that period. This will be used to check for wave presence off-zenith and test its validity by the phase tilt's linearity. The new version of the text brings this description to section 2.2.1.

In Figure 4, there are vertical red lines indicating the selected lines. Are these lines inserted manually? Those lines are placed at distinct spots in the zonal and meridional keograms. How is their placement?

They are automatically selected in the time domain, half a period before and after the central observation time. In the vertical (spatial domain), they were selected based on the quality of the linear fitting; or in other words, the flattest region.

In Figure 5, what is the meaning of the red phase lines? The figure legend does not inform it. Additionally, the vertical axis lacks units, which I suppose should be kilometers.

The units are degrees. The red part of the phase lines is the sector used for the coeficient estimations, selected automatically by the program based on the quality of the fitting. The figure and legend have been updated to make it clear.

Regarding the two synthetic waves used to test the method. Do they have the same amplitude? The wavelet spectrum in Figure 7 shows distinct power for the waves.

They have the same amplitude, and so does the noise. The difference in the power spectrum is due to the wavelet properties that scatter the power density. This is corrected by the amplitude correction indicated in section 2.3.

Figure 8, what are the units of the vertical axis?

The unit is degree. Figure was updated to make it clear.

Authors state that they used "more than 800 keograms to run the tests of method". How were those keograms generated? Do the synthetic waves have the same properties (e.g. wavelengths, periods, amplitudes)? Explain the distribution of the wave parameters.

The new version of the text (lines 258-265) details the set of simulations.

At page 11, lines 191-196, the authors state some issues related to the amplitude of the synthetic waves retrieved through their method. The main reason is the superposition of the waves. In the atmosphere, waves do coexist, making this an issue that is difficult to avoid.

The authors agree with this. The new text version discusses this point more in the errors section. We also expect our manuscript to provide more insights on how to lessen this issue, as the method to validate the phase lines indicated an important feature of the wave superposition. Refer to lines 266-273 of Section 4.

On page 12, lines 215-218, the authors suggest that the direction of propagation is isotropic, but say that the waves propagate preferentially eastward. It seems contradictory. Additionally, state that there is no filtering pattern. I infer they are referring to filtering by background wind. It is important to make this clear. Moreover, explain why such filtering is not observed for these waves.

In our research field, we typically seek anisotropies associated with background wind filtering patterns (see, e.g., Giongo et al., 2020), and that is what we meant there and with the presented figure. In addition to MSGW's present anisotropy, where most of the waves propagate eastward, we could not determine whether the eastward propagating waves are the faster ones, which indicates wind filtering anisotropy. No wind filtering is detected because the waves are much faster than the wind, allowing them to propagate from the ground without being absorbed by the background.

On page 14, line 240-244. The expression "Above the equatorial region" is used a couple of times referring to observations at Cariri (7.4º S). This phrasing sounds unusual.

The paragraph was modified to make the text more clear.

On page 15, lines 250-255. Amplitudes of gravity waves inferred from airglow are rare because the all-sky imagers must be calibrated to measure absolute intensity.

That is indeed a current problem. The imagers have not been calibrated to show the airglow brightness in an appropriate unit. Therefore, we must explore the wave amplitudes in terms of relative brightness and somehow convert this to temperature (a procedure being explored in the next paper composing the first author's thesis)

The authors analyzed gravity waves observed in the Antarctic Peninsula and compared them with waves observed in the equatorial region. For the purpose of the paper, which I assume is to validate their methodology, this comparison may not be ideal.

The comparison aimed to discuss the results with other observation sites, a common practice in our research field. The discussions were revised in the current version.

The keogram technique is widely used to investigate medium-scale gravity waves in airglow images. Since the authors are proposing an automatic algorithm, they should validate their results against the standard keogram analysis by manually analyzing the wave parameters.

The authors included in the discussion section a comparison with the older methodology along with the key advancements achieved through the newly developed methodology. Unfortunately, standard analysis methods are not recognized in the literature. The authors are pleased to demonstrate that their method can serve as a reference by showing the error estimation in relation to the phase fitting quality. This new discussion can be found in lines 331-351.

**Minor**
page 3, line 63
63ºC degree -> 63ºC
page 3, line 79
Fig 1a -> Figure 1a
page 3, line 87
disposing of them -> disposing them
Maybe the word "disposing" is not the best choice to describe the situation. It could be "arranging
them", "stacking them".
page 3, lines 87-88
shorter periods representing high-frequency waves easily analyzed by other methods -> shorter
periods representing high-frequency waves can be easily analyzed by other methods

page 4, lines 105-106
Prominent peaks can be seen near 20, 30, 50, 80, and 140 minutes of period ->Prominent peaks can be seen at periods near 20, 30, 50, 80, and 140 minutes

page, 5, line 118 average around ->averaged around

page 12, line 212, 20 min was the lesser value ->20 min was the lowest value

The authors thank the Referee for the minor suggestions.

---

## Author Comment (AC2)

Authors' response to Referee 2:
Gabriel Giongo et al.

The authors thank the referee for reviewing the paper. All comments are relevant, and the suggestions are very useful in improving the final version of the paper. Replies to all the comments are addressed below.
Reviewer comments in black, author's comments in blue.

**Overview**

This manuscript introduces an improved method for detecting medium-scale gravity waves from ground-based airglow imagers. The method is based on keogram analysis, it is designed to automatically detect wave events and determine the horizontal wavelength, azimuth, period and ground-based phase velocity of the waves. It was tested using simulated data as well as actual airglow images. Authors report reliable wave identification and low errors of the retrieved wave parameters.

Unfortunately, the new methods, although they seem to contain good ideas for dealing with many aspects of the problem, are not explained clearly enough, to the point that I had difficulties interpreting some of the main results (see major comments below). I have also outlined some possible issues with the method itself and made a few suggestions, I hope they can be of some use. Since introduction of the improved methodology is the main goal of this paper, I cannot recommend it for publication in its current state.

In general, we modified much of the text, completely rewrote many parts, and added a subsection for an overview of the methodology and the mathematical concepts and physical meanings associated with them. This overview is provided before demonstrating performance in synthetic images and then in the real keograms, highlighting the problems and discussing them. Additionally, the error retrieval by the simulation methods is in its own section, and the discussion section presents results from the previous methodology and more clearly points out the improvements brought by the newly developed methodology. Furthermore, we proposed a new title to emphasize the paper's contribution to the current spectral analysis of gravity waves. The figures have been enhanced with clearer plots and legends.

**General/major comments**

**1. Severe lack of clarity in mathematical methodology and result presentation.** This manuscript cites quite a few previous works in relation to various methods used in wave analysis and error estimation. While it is the accepted norm for derivations of those results and their finer detail, the authors of this manuscript rely solely on the references and very general methodological concepts (e.g. "error propagation") to describe the key quantitative parameters and mathematical methods used in their work. Because of this, I had difficulty understanding which physical or statistical quantities are described by the provided numerical values. Some of the most important results are presented in figures with undefined color scales, or as a percentage of an unstated quantity or as a standard deviation, without a clear a description of the data set for which the standard deviation was calculated (see major comment 2 and specific comments for concrete examples). I must apologize if I overlooked something or if some of the subsequent comments are simply a result of me interpreting some physical quantity not in the way that the authors intended, but, in my opinion, reader should not need to guess quite so often.

In the new version of the text, all mathematical concepts and the presentation of results were rewritten for improved clarity. Figures, along with their legends and tables, were updated. More details were added to the methods and the rationale behind their use. In the new version, we aimed to enhance clarity even for those not in the field.

**2. Simulated keogram analysis results.** The authors mention "error propagation" in several places in the manuscript. For example, they state that "It is important to emphasize that this study is not trying to validate the wavelet transform or other well-known mathematical procedures, like linear fitting. Still, it tracks errors throughout the combination of processes employed. Errors due to any transform or mathematical procedure exist and can be studied or found in other publications (Liu et al., 2007; Xu et al., 2024)." This leads me to believe that quantities described as "propagated errors" are simply a result of statistical errors of well-know data analysis procedures (e.g. linear fitting, wavelet analysis) combined using the standard methods of calculating the error of a derived quantity from the errors of the parameters the quantity was derived from. On the other hand, running the analysis on simulated data provides an independent method of estimating the error: one can simply compare the results of analysis to the known parameters of the simulated data. Therefore:

(a) I do not understand what authors mean in the beginning of Section 3, when they state that "Artificial images made as arrays containing known wave patterns were used to verify the quality of the analysis, track the error propagation through the procedure, and estimate future errors generated throughout the process". Does this mean that the error estimates for real data run (i.e. "propagated error") was somehow based on the results of the simulated data run? If yes, then how? Was the simulated data used only for setting the threshold for acceptable linear phase fits, or for something else?

The simulations are used for error estimation due to the procedure, so what we call error propagation is simply the difference between the results and the input values. They also provided insights into how errors are generated throughout the procedure and how the superposition of waves can influence these errors. We acknowledge the mistake here; for the real keograms, the deviations mentioned in Table 1 reflect the standard deviations of the parameters estimated as an average of multiple phase line selections. For simulations, only one phase line is necessary. As the new text alters the order of the descriptions, we now mention that we take an average parameter in the real keograms from the phase lines, as it is essential to monitor the wave presence over time. Section 3 (lines 219-248) is an updated text with the description of the average parameters for real keograms, and Section 4 (lines 250-294) is for the latest description of error propagation throughout the procedure.

(b) Authors only provide one example of a simulated keogram, while "more than 800" were used in total. More information on this keogram set should be provided. Did every simulated keogram have two waves in it? What were the wavelengths and periods? What were the relative amplitudes?

The new text version provided details about the data set used in the simulations, including wavelengths, periods, and amplitudes (please see lines 258-265). The consequences of the waves' superposition on the phase signal were also demonstrated to illustrate the problems better and facilitate further discussions (lines 266-273).

(c) (Important!) Table 2 is extremely confusing. Firstly, I have no idea what the "standard deviation" means here. I did not find an explanation anywhere in the main text. Variance of the linear fit for phase is discussed in detail in this section, but it is not clear how it would be

used to obtain the standard deviations for all the measured parameters. Are these just the standard deviations of the parameters used for all the different simulated waves? Secondly, what kind of error estimates are given in the table? Since this is a test on simulated data, it is vital to provide a clear overview of how wave analysis results compare to the actual parameters of simulated waves. Yet the table suggests that this is "propagated error". If this is just the propagated error in the sense I defined it at the beginning of this comment, what was the point of the simulated data run and why are the authors not showing how the analysis results compare to the actual wave parameters? If Table 2 does indeed show this comparison, then how do these errors compare to the errors provided for the real keograms (which are the "propagated errors", right?), where the wave parameters were not known in advance?

Table 2 summarizes Figure 9 (old version). It displays the distribution of errors as a function of variance. We calculated the mean value and standard deviation of this mean. Every distribution can have a mean, standard deviation, skewness, and kurtosis, which are referred to as statistical moments. We summarize the simulation results in the table: errors throughout the procedure and the threshold used to limit those errors. This emphasizes the low error propagation in the methodology, alongside other errors that may arise from actual keograms, which are discussed adequately in the new text version. We hope the new text version clarifies the statistical concepts and the discussion regarding error estimations and parameter errors.

**3. Collocation of zonal and meridional wavelength estimates.**

(a) As one can see from figures 4 and 5, the wavelengths can end up being retrieved from a part of keogram that does not include the center (zenith) pixel. This can lead to a situation when zonal and meridional wavelengths are retrieved from completely disjoint parts of the image, neither of which contain the zenith pixel (which is used to identify temporal peaks in wave activity). In my opinion, this a major flaw of the method. Many structures seen, for example, in Figure 4, do not span the whole spatial extent of the keogram: there are major differences in phase structure and phase tilt between North and South, as well as East and West, parts of the image. Therefore, I would assume that method would often estimate the meridional and zonal wavelengths of completely different wave packets. The authors also seem to be aware of this: they state in Section 5 that waves not spanning entire keogram are a problem and that "The resulting period can be closer to one wave while the resulting wavelength is closer to another". It therefore hard for me to understand why so little care is taken to ensure the collocation of both wavelength estimations and the wave activity peak identification. Point (b) outlines one possible way to resolve that, just as an example.

This is indeed a significant issue concerning the keogram technique, and many previous analytical methods have failed to address the problem. Since the earlier methods were manually oriented, the operator had to be cautious when selecting the same wave packet in both components of the keogram. Our central assumption is that the wave has a large scale, allowing it to traverse a significant portion of the images and persist over time. Small-scale waves more frequently pass through only a portion of the image, while it is rarer for medium-scale waves to do so. In our method, we could avoid the "non-zenith" waves by preventing the program from selecting the best-fitting region near the edges of the keogram. However, we chose not to do this and instead rely on further validation through human approval, which is one reason we maintained visual validation (Major comment 4). Differences in phase tilt between components are expected, as the inclination is proportional to the wavelength projected in that direction. It is entirely possible that the wave lacks a component in that

direction. The potential for a wave to have the parameters of another was observed in the simulations and is shown in more detail in lines 266-273.

(b) As far as I understand, the method described in the manuscript only makes use of one row and one column from each image. Furthermore, the peaks of wave activity are only identified from a time series obtained from a single pixel. It seems to me, that a simple way to improve the method would be to pick a few (let us denote their number by n) equally spaced rows and the same number equally spaced columns in the image. Then one would apply the wavelet power spectrum analysis not just for the zenith pixel, but for the n2 intersections of rows and columns. The strongest peaks in these power spectra could then be used to identify wave events, and wavelengths could be estimated from the meridional and zonal keograms that intersect at the point with a strong power peak. This way, wave events that happen off-zenith could be identified. Even more importantly, one could apply stricter colocation criteria for phase fitting (e.g. only use the parts of the two keograms that contain their intersection point, where the power spectrum peak was identified) without loosing too many wave events. Finally, if this method ends up detecting waves in close spatial and temporal proximity to one another, one could use this as a consistency check. As far as I understand, none of the methods described in this manuscript are very computationally expensive, so it should not be a problem to run this for n = 5, for example. There could be, of course, many more (and probably better) ways to resolve the colocation problems and make better use of data from the whole image.

You understood right. Your suggestion is good and we thank you for trying to help us. Our group already did the keograms made with the lines (and columns) out of the center, and no practical consequences were verified. Also, we tried the cross-spectra of the Fourier (and wavelet) from lines in a similar way you suggested. Cross-spectra indicate the wave modes present in both spectra, so we do not need to check "the strongest peaks in these power spectra", as the spectrum does not show power if the mode is absent in one of the series. Other problems arise, like the Milky Way signal breaking the wave modes as it is stronger and present everywhere in the keogram; and we have doubled results for the same wave, making it much worse to check after the end of a procedure or add another ad hoc parameter. As in the previous point, we highlight that the waves are large and slow enough to pass through the zenith for a considerable time to be analyzed reliably by keograms. Discussion on this point was inserted in the text and can be checked at lines 352-357.

**4. The need for manual intervention.** In the beginning of Section 2.2 the authors state "Although the procedure is based on Fourier Transform as in previous works, the new feature of this methodology is that it automatically selects the waves along the keogram and verifies the quality of the oscillatory signal, resulting in less user bias on its usage". Therefore, the automatic nature of the new analysis method is the key novel aspect in this work. However, Section 5 implies that some of the most important flaws of the method could be adressed by manual inspection of results. In particular, phase lines of valid wave fits "can be bent but not tortuous, which is a consequence of a close superposition of the waves.". In my understanding, distinguishing a "tortuous" phase line from a bent one should not be too difficult to accomplish automatically: one could consider, for example, checking whether some norm (e.g. the Euclidean norm) of the second temporal derivative of the phase line exceeds a threshold. That would, of course, introduce another ad-hoc parameter to the analysis (besides the currently used threshold for the variance of linear fit), but this would still be more objective than manual inspection of results, that the authors have set out to avoid. Also, was any manual selection of wave fits used to obtain any of the results shown in the manuscript?

All the procedures are fully automated; we simply conduct a visual check of the procedure's performance by observing the reconstruction and the phase lines after the procedure concludes, in order to approve or deny the wave analysis result. The final inspection ensures that it is a wave. We believe it is unwise to use the results from extensive data sets, which retrieve numerous wave parameters, without validating the results. We have enhanced the discussion section on this matter; the user now receives the best results and quality automatically, without the need to spend time selecting the area where the wave is, relying solely on their own judgment for the analysis. Furthermore, while we could utilize a machine learning-based program to automatically select the waves, this depends on large data sets of validated results that do not yet exist, primarily due to the limited number of medium-scale gravity wave analyses conducted so far. (ML programs have been employed to identify clear sky nights, for which we have extensive validated observations, and they are quite helpful). The suggestion regarding the Euclidean norm is quite good, and we may implement it in the future; for now, the program has demonstrated excellent performance in its current state and represents a significant improvement in keograms analysis. Introducing another ad hoc parameter is not our intention, as we would need to repeat all the tests conducted in this work and lack a consolidated physical meaning for it, unlike the linearity of the phase lines. We refer to lines 134-138, where we explain the assumption made, which is further supported by Figure 10.

**5. Validation against whole images.** Table 2 shows the parameters of some retrieved waves. Those waves have wavelengths that are generally similar, or smaller, than the spatial extent of the airglow image (512 × 512 km). Would it not be possible to verify some of the image parameters obtained from keograms against the corresponding images? I understand that some of the images would be affected by the Milky Way and various other light sources, but would it not be possible to obtain at least some images, where the medium-scale waves discussed would be visible (perhaps after applying low-pass filter to suppress small scale structure)? I agree that keogram analysis has many advantages over this approach, but could individual images be useful at least for demonstrating that wave azimuth and wavelength were retrieved correctly? If this is indeed very difficult, maybe authors could have validated their new method against previous, more manual keogram-based wave analysis techniques?

The new version includes discussions, a comparison with the older method, and a highlight of how this new method has improved spectral analysis. The discussion is in lines 331-351.

**Minor/specific comments**
1. L29: I do not understand this sentence. The geographic location and altitude of a particular satellite observation (which is typically meant by the term "geolocation") can typically be determined pretty accurately. In particular, nadir-viewing satellite instruments (e.g. AIRS, AWE missions), offer similar geolocation accuracy as ground-based airglow imagers. Maybe the authors are talking about the spatial data coverage, and not geolocation here? Also, "local features and properties" is an extremely vague description given that methodology is the main topic of this paper! Authors should explain this statement in a lot more detail or remove it.

The paragraph was rewritten to clarify the argument that motivates the work (lines 27-37).

2. L106: "The peak identification procedure needs an equally dimensioned array". Why is that? Is that a fundamental requirement of the method, or is this just needed for compatibility with some particular implementation, software library, etc.?

It is necessary for our particular procedure for peak estimation. Details of the procedure were included in the text (lines 181-191), as also required by Referee 1.

3. Figure 2: The color scale in this figure is not defined or explained anywhere. This must be fixed.

It is a simple grayscale weighted by the maximum and minimum values of the entire image. Since the airglow images are not calibrated, they depend on an arbitrary basis based on the integration time. The figure legend has been updated.

4. Figure 3: Percentage of what? Power spectrum is normalized with respect to what exactly?

Concerning the maximum value. The figure legend was updated.

5. Figure 4: Units for deviation are not given. Also, the choice of color scale is very unfortunate, as there is very little contrast for deviations between 0 and 15, hence only the negative phases of most waves are visible.

The image was completely rescaled to fit the detected amplitude of the wave. Unfortunately, this caused other signals to become saturated. We simply need to choose which one will be saturated, as the Milky Way is the strongest feature in airglow images.

6. Equation (5): while it is, of course, OK to refer the reader to the study by Xu for derivation of this equation, authors should explain clearly what Cf is and how it was used in this work.

The text was revised to include additional details (lines 209-211).

7. L216: "Waves for a broad spectrum of velocity propagate in all directions without apparent anisotropy. However, more waves are seen with zonal components preferentially in the eastward direction compared to the westward direction". In my understanding, the second sentence directly contradicts the first: more waves with zonal components in the eastward direction is anisotropy.

In our research field, we typically look for anisotropies related to background wind filtering patterns (see, e.g., Giongo et al., 2020), which is what we referred to there and in the presented figure. Besides the MSGW's present anisotropy, with most of the waves propagating to the east, we could not determine whether the eastward-propagating waves are the faster waves, which constitutes the wind filtering anisotropy. No wind filtering is detected because the waves travel much faster than the wind, so they could have propagated from the ground without being absorbed by the background.

**Minor typos and suggestions**
This is a list of typos that I noticed and minor, mostly language related, suggestions. Point-by-point replies to these are not necessary.

[...]

The authors thank the Referee for the suggestions.